# Independent Risk Factors Predicting Eradication Failure of Hybrid Therapy for the First-Line Treatment of *Helicobacter pylori* Infection

**DOI:** 10.3390/microorganisms12010006

**Published:** 2023-12-19

**Authors:** Chien-Lin Chen, I-Ting Wu, Deng-Chyang Wu, Wei-Yi Lei, Feng-Woei Tsay, Seng-Kee Chuah, Kuan-Yang Chen, Jyh-Chin Yang, Yu-Hwa Liu, Chao-Hung Kuo, Sz-Iuan Shiu, Chang-Bih Shie, Kuan-Hua Lin, Chia-Long Lee, Ping-I Hsu

**Affiliations:** 1Department of Medicine, Buddhist Tzu Chi General Hospital, Tzu Chi University, Hualien 970, Taiwan; clchen@tzuchi.com.tw (C.-L.C.); aquarious@seed.net.tw (W.-Y.L.); 2Division of Gastroenterology, Department of Medicine, An Nan Hospital, China Medical University, Tainan 709, Taiwancbshie@gmail.com (C.-B.S.);; 3Division of Gastroenterology, Department of Internal Medicine, Kaohsiung Medical University Hospital, Kaohsiung Medical University, Kaohsiung 807, Taiwan; dechwu@yahoo.com (D.-C.W.); kjh88kmu@gmail.com (C.-H.K.); 4Division of Gastroenterology and Hepatology, Department of Internal Medicine, Kaohsiung Veterans General Hospital, National Yang-Ming University, Kaohsiung 813, Taiwan; fwchaie@vghks.gov.tw; 5Division of Gastroenterology, Department of Internal Medicine, Kaohsiung Chang Gung Memorial Hospital, College of Medicine, Chang Gung University, Taoyuan 333, Taiwan; chuahsk@seed.net.tw; 6Division of Gastroenterology and Hepatology, Department of Internal Medicine, Taipei City Hospital, Renai Branch, Taipei 106, Taiwan; 7Division of Gastroenterology, Department of Internal Medicine, National Taiwan University Hospital, Taipei 100, Taiwan; jcyang47@ntu.edu.tw; 8Division of Gastroenterology, Department of Internal Medicine, Shin Kong Wu Huo-Shih Memorial Hospital, Taipei 111, Taiwan; skh0724@yahoo.com.tw; 9Division of Gastroenterology and Hepatology, Department of Internal Medicine, Taichung Veterans General Hospital, Taichung 407, Taiwan; b9002007@hotmail.com; 10Division of Gastroenterology and Hepatology, Department of Internal Medicine Cathay General Hospital, Taipei 106, Taiwan

**Keywords:** *Helicobacter pylori*, hybrid therapy, risk factors, eradication failure, resistance

## Abstract

Hybrid therapy is a recommended first-line anti-*H. pylori* treatment option in the American College of Gastroenterology guidelines, the Bangkok Consensus Report on *H. pylori* management, and the Taiwan *H. pylori* Consensus Report. However, the cure rates of eradication therapy in some countries are suboptimal, and the factors affecting the treatment efficacy of hybrid therapy remain unclear. The aim of this study is to identify the independent risk factors predicting eradication failure of hybrid therapy in the first-line treatment of *H. pylori* infection. A retrospective cohort study was conducted on 589 *H. pylori*-infected patients who received 14-day hybrid therapy between September 2008 and December 2021 in ten hospitals in Taiwan. The patients received a hybrid therapy containing a dual regimen with a proton pump inhibitor (PPI) plus amoxicillin for an initial 7 days and a quadruple regimen with a PPI plus amoxicillin, metronidazole and clarithromycin for a final 7 days. Post-treatment *H. pylori* status was assessed at least 4 weeks after completion of treatment. The relationships between eradication rate and 13 host and bacterial factors were investigated via univariate and multivariate analyses. In total, 589 patients infected with *H. pylori* infection were included in the study. The eradication rates of hybrid therapy were determined as 93.0% (95% confidence interval (CI): 90.9–95.1%), 94.4% (95% CI: 93.8–97.2%) and 95.5%% (95% CI: 93.8–97.2%) by intention-to-treat, modified intention-to-treat and per-protocol analyses, respectively. Univariate analysis showed that the eradication rate of clarithromycin-resistant strains was lower than that of clarithromcyin-susceptible strains (83.3% (45/54) vs. 97.6%% (280/287); *p* < 0.001). Subjects with poor drug adherence had a lower cure rate than those with good adherence (73.3% (11/15) vs. 95.5% (534/559); *p* = 0.005). Other factors such as smoking, alcohol drinking, coffee consumption, tea consumption and type of PPI were not significantly associated with cure rate. Multivariate analysis revealed that clarithromcyin resistance of *H. pylori* and poor drug adherence were independent risk factors related to eradication failure of hybrid therapy with odds ratios of 4.8 (95% CI: 1.5 to 16.1; *p* = 0.009) and 8.2 (95% CI: 1.5 to 43.5; *p* = 0.013), respectively. A 14-day hybrid therapy has a high eradication rate for *H. pylori* infection in Taiwan, while clarithromycin resistance of *H. pylori* and poor drug adherence are independent risk factors predicting eradication failure of hybrid therapy.

## 1. Introduction

*Helicobacter pylori* (*H. pylori*) infection is an important pathogen for the development of chronic gastritis, duodenal ulcer, gastric ulcer, gastric adenocarcinoma and mucosa-associated lymphoid tissue lymphoma [1,2,3,4]. Eradication of *H. pylori* has also been shown to prevent the recurrence of peptic ulcer disease [5,6] and cure gastric mucosa-associated lymphoid tissue lymphoma [7,8]. Additionally, *H. pylori* eradication is recommended as a preventative measure in regions with a high incidence of gastric cancer [9,10].

Most international guidelines recommend standard triple therapy containing a proton-pump inhibitor (PPI), clarithromycin and amoxicillin (or metronidazole) for 14 days as the choice of treatment for eradicating *H. pylori*, especially in regions with low clarithromycin resistance. With the increasing prevalence of antibiotic resistance of *H. pylori*, 7-day standard triple therapy is no longer the recommended option for treating *H. pylori* infection [11,12,13,14,15]. Recently, several strategies including bismuth quadruple therapy [16,17], non-bismuth quadruple therapy (e.g., concomitant therapy, hybrid therapy and sequential therapy) [18,19,20,21,22], high-dose dual therapy [23] and vonoprazan-containing therapies [24,25] have been proposed to increase the eradication outcome.

According to the Maastricht V/Florence Consensus Report [26], bismuth quadruple therapy is the preferred first-line treatment for *H. pylori* infection in areas with high clarithromycin resistance (>15%) and an alternative in areas with low clarithromycin resistance. However, the administration of bismuth quadruple therapy is complicated, and frequency of adverse effects are extremely high (47% to 67%) [27,28,29]. These shortcomings easily reduce the adherence of patients.

High-dose dual therapy is a new treatment in the first-line treatment of *H. pylori* infection [30]. This therapy involves high-dose PPI that maintains the intragastric pH at a value higher than 6.5 [31]. High-dose amoxicillin in the regimen can keep a high plasma concentration of amoxicillin (>minimal inhibitory concentration) for eradicating *H. pylori* [32]. This novel modality is simple to administer. Additionally, its frequency of adverse events was lower than that of standard triple therapy [33]. A randomized controlled trial from China demonstrated that 14-day high-dose esomeprazole–amoxicillin dual therapy and 14-day bismuth-containing quadruple therapy had comparable eradication rates in the first-line treatment of *H. pylori* infection (intention-to-treat analysis: 90% vs. 87%) [34]. However, this novel treatment was less effective for *H. pylori* eradication in the United States [23] and Korea [35]. In addition, a recent multi-center, randomized controlled trial showed that high-dose dual therapy was inferior to hybrid therapy in the first-line treatment of *H. pylori* infection [36].

Hybrid therapy contains a dual regimen with a PPI plus amoxicillin for an initial 7 days followed by a quadruple regimen with a PPI plus amoxicillin, metronidazole and clarithromycin for a final 7 days [20]. In Taiwan, hybrid therapy and bismuth quadruple therapy had comparable efficacy in eradicating *H. pylori* infection in the first-line treatment [36,37]. In Iran, 14-day hybrid therapy was superior to 14-day standard triple therapy for *H. pylori* eradication [38]. A randomized controlled trial by Sardarian et al. also demonstrated that 14-day hybrid therapy achieved a higher eradication rate than 10-day sequential therapy [39]. Furthermore, a large multicenter randomized controlled study from Spain and Italy reported that the eradication rates of both 14-day hybrid and 14-day concomitant therapies were more than 90% for the first-line treatment of *H. pylori* infections in areas with high clarithromycin and metronidazole resistance. There were no differences in cure rates between the two therapies [40]. According to the American College of Gastroenterology (ACG) guidelines on the treatment of *H. pylori* infection, 14-day hybrid therapy is currently recommended as one of the first-line therapies [41]. In the Bangkok Consensus Report on *H. pylori* management in the ASEAN, hybrid therapy is also a recommended option in the first-line treatment of *H. pylori* infection [42].

Hybrid therapy can achieve a high eradication rate in Taiwan, Iran, Greece, Spain and Italy, but its eradication rates for *H. pylori* infection in China and Korea are suboptimal [43,44]. Although clarithromycin resistance has been identified as a bacterial factor affecting the success of hybrid therapy [36], whether host factors such as smoking, alcohol drinking and drug adherence can influence the cure rate remain unclear. To investigate the independent risk factors predicting eradication failure of 14-day hybrid therapy for *H. pylori* infection, we conducted a retrospective cohort study.

## 2. Methods

### 2.1. Subjects

A retrospective cohort study was conducted on *H. pylori*-infected patients who underwent 14-day hybrid therapy between September 2008 and December 2021 in ten hospitals across Taiwan, including the Tainan Municipal An Nan Hospital, Kaohsiung Veterans General Hospital, Kaohsiung Medical University Hospital, Kaohsiung Chang Gung Memorial Hospital, the Renai Brach of Taipei City Hospital, National Taiwan University Hospital, Shin Kong Wu Ho-Su Memorial Hospital, Buddhist Tzu Chi General Hospital, Cathay General Hospital and Taichung Veterans General Hospital. The exclusion criteria included (1) allergy to eradication medications used in the study, (2) history of gastrectomy, (3) history of previous eradication therapy, (4) pregnancy, (5) lactation, (6) use of antibiotics within 4 weeks before treatment and (7) coexistence of severe illness. The study was conducted according to a standard protocol in accordance with the principles of good clinical practice from the Declaration of Helsinki. All the authors had access to the study data and had reviewed and approved the final manuscript.

### 2.2. Design

During the study period, patients infected with *H. pylori* were administered a 14-day hybrid therapy. The therapy consisted of a dual regimen with a PPI plus amoxicillin for an initial 7 days and a quadruple regimen with a PPI plus amoxicillin, metronidazole and clarithromycin for a final 7 days. Trained assistants prospectively followed up with all patients using a standard protocol [20]. They were asked to return in the second week to assess drug adherence and adverse effects. Post-treatment *H. pylori* status was assessed at least 4 weeks after completion of treatment using endoscopy with histology and rapid urease test or a ¹³C-urea breath test. PPIs and histamine-2 receptor antagonists were prohibited for at least 2 weeks before post-treatment *H. pylori* assessment. ^13^C-urea breath test with the *Proto Pylori kit* (Isodiagnostika, Montreal, QC, *Canada*) was conducted after an overnight fast. An infrared spectrometer was used to assess baseline and 30 min breath samples. Positive results were defined as δ^13^CO_2_ value ≥ 4 units. Negative results were defined as δ^13^CO_2_ value < 2.5 units. Subjects having inconclusive data underwent another ^13^C-urea breath test at least 4 weeks later until the results of urea breath test became conclusive. Technicians carrying out the tests were blinded to the therapies received by patients. Cure of *H. pylori* infection was defined as (1) negative results of both histology and rapid urease test or (2) a negative result of urea breath test.

*H. pylori* infection status was determined by rapid urease test (*n* = 570), histology (*n* = 567) and/or culture (*n* = 350) before treatment. Only patients with positive results of at least two tests were included for the study. Patients were instructed to fill out a special questionnaire detailing age, sex, medical history, smoking status, history of alcohol drinking and coffee and tea consumption. Smoking was defined as consumption of cigarettes ≧ 1 pack/week. Alcohol consumption was defined as alcohol drinking > 80 g/day. Coffee or tea consumption was defined as drinking ≧ 1 cup/day. We informed the patients of the common adverse events of eradication therapies before treatment. They were asked to record adverse events in provided diaries. The degree of adverse events was assessed by a four-point scale system: none, mild (discomfort that is annoying but does not interfere with daily life), moderate (discomfort that interferes with daily life) and severe (discomfort that results in discontinuation of eradication therapy). Drug adherence was checked by counting the number of unused medications. Poor drug adherence was defined as taking less than 80% of total medicines used in eradication regimens.

We used Pronto Dry^®^ kit (Medical Instruments Corp., Brignais, France) for rapid urease test. Two biopsy specimens taken from the lesser curvature sites of the antrum and the corpus were used for histological examination. *H. pylori* culture was conducted according to previously described methods [36]. Gastric specimens were cultured on Brucella chocolate agar with 7% sheep blood and incubated for 7 days under microaerobic conditions. Antibiotic susceptibility was assessed using the E-test (AB Biodisk, Solna, Sweden). *H. pylori* strains with minimal inhibitory concentration values (MIC) > 0.5 μg/mL, >1 μg/mL, >8 μg/mL and >4 μg/mL were considered to be resistant to amoxicillin, clarithromycin, metronidazole and tetracycline, respectively [36].

### 2.3. Statistical Analysis

The analysis was conducted using intention-to-treat, modified intention-to-treat and per-protocol analyses. Intention-to-treat analysis included all randomized patients who took at least one dose of the eradication medicines. Patients who did not receive post-eradication *H. pylori* assessment were considered treatment failures in the intention-to-treat analysis. The modified intention-to-treat population included all randomized patients except the patients whose post-eradication *H. pylori* status was unknown. The per-protocol analysis excluded the patients who violated the protocol, such as patients who did not take at least 80% of the treatment drugs, and those who did not receive post-treatment *H. pylori* assessment.

To identify the factors affecting the treatment response, clinical and endoscopic parameters were analyzed using univariate analysis for the modified intention-to-treat population. We compared categorical data using the χ^2^ test or Fisher’s exact test, as appropriate. Continuous data were compared by Student t test. All statistical analyses were conducted using SPSS (version 12.0 for Microsoft Windows). The clinical and bacterial factors analyzed in the study included age (<60 or ≥60 yr), gender, smoking, alcohol consumption, ingestion of coffee, ingestion of tea, endoscopic finding (peptic ulcer or gastritis), antibiotic resistance, type of PPI and drug adherence. The factors with a *p* value < 0.05 in the univariate analysis were further assessed using multivariate analysis to identify the independent risk factors predicting treatment failure. A *p* value < 0.05 was considered statistically significant.

## 3. Results

### 3.1. Baseline Characteristics

A total of 589 *H. pylori*-infected outpatients were enrolled in this study. The baseline demographic and clinical characteristics of the patients are summarized in Table 1. The mean age of the patients was 54.5 ± 11.9 years. The frequencies of antibiotic resistances for clarithromcyin, amoxicillin, metronidazole and tetracycline were 15.4%, 0.3%, 38.3% and 0%, respectively. Fifteen patients were lost to follow-up after treatment and excluded from risk factor analysis for eradication failure.

### 3.2. Eradication Outcomes

The outcomes of the 14-day hybrid therapy are listed in Table 2. The eradication rates of hybrid therapy were 93.0% (95% confidence interval (CI): 90.9–95.1%), 94.4% (95% CI: 93.8–97.2%) and 95.5%% (95% CI: 93.8–97.2%) as determined by intention-to-treat, modified intention-to-treat and per-protocol analyses, respectively. The frequency of adverse events of hybrid therapy was 21.1% (124/589; 95% CI: 17.8–24.4%). The drug adherence rate was 97.6% (575/589; 95% CI: 96.4–98.8%).

### 3.3. Univariate Analysis for Risk Factors Related to Eradication Failure of Hybrid Therapy for H. pylori Infection

Table 3 shows the univariate analysis data for parameters influencing eradication rates of the 14-day hybrid therapy in the modified intention-to-treat analysis. Using 1 μg/mL as the MIC breakpoint of clarithromycin, the eradication rate in the patients harboring clarithromycin-resistant strains was lower than that in those harboring clarithromcyin-susceptible strains (83.3% (45/54) vs. 97.6%% (280/287); *p* < 0.001). In addition, patients with poor adherence had a lower eradication rate than those with good adherence (73.3% (11/15) vs. 95.5% (534/559); *p* = 0.005). The other factors, including smoking, alcohol drinking, coffee consumption, tea consumption, gender, age and type of PPI, were not significantly associated with the cure rate.

### 3.4. Independent Risk Factors Predicting Eradication Failure of Hybrid Therapy for H. pylori Infection

Table 4 demonstrates the results of multivariate analysis for the independent risk factors predicting eradication failure of 14-day hybrid therapy for *H. pylori* infection. The data revealed that clarithromcyin resistance of *H. pylori* and poor drug adherence were independent risk factors related to eradication failure of hybrid therapy with odds ratios of 4.8 (95% CI: 1.5 to 16.1; *p* = 0.009) and 8.2 (95% CI: 1.5 to 43.5; *p* = 0.013), respectively.

## 4. Discussion

In this large retrospective cohort study, we included 589 *H. pylori*-infected patients who received 14-day hybrid therapy, and identified the independent risk factors for eradication failure by this treatment as the first-line therapy for *H. pylori* infection. The data showed that the success rates of hybrid therapy determined by the intention-to-treat, modified intention-to-treat and per-protocol analyses were 93.0%, 94.4% and 95.5%, respectively. Multivariate analysis revealed that clarithromycin resistance of *H. pylori* and poor drug adherence of the host were independent risk factors for the eradication failure of hybrid therapy, with odds ratios of 4.8 and 8.2, respectively.

In the current study, the eradication rate in the patients harboring clarithromycin-resistant strains was lower than that in those harboring clarithromcyin-susceptible strains (83.3% vs. 97.6%). These findings are consistent with another clinical trial that demonstrated clarithromycin resistance of *H. pylori* as a risk factor for eradication failure of hybrid therapy in China [43]. In that study, the eradication rates of hybrid therapy for clarithromycin-resistant and susceptible strains were 74.6% and 91.7%, respectively. Clarithromycin resistance of *H. pylori* significantly affected eradication outcome. However, another study from Taiwan did not demonstrate clarithromycin resistance of *H. pylori* as a risk factor for eradication failure by hybrid therapy [45]. The variable results of hybrid therapy for *H. pylori* treatment may be due to the differences in the potencies of PPIs, *CYP2C19* genotypes, and resistance profiles of metronidazole and amoxicillin.

Poor drug adherence has been reported as an independent risk factor of eradication failure for standard triple therapy [46] and high-dose dual therapy [36]. In this study, poor drug adherence was also identified as an independent risk factor predicting eradication failure of hybrid therapy. The eradication rate in patients with poor drug adherence was lower than that in those with good drug adherence (73.3% vs. 95.5%). The data suggest that adequate patient education on drug adherence is important in the treatment of *H. pylori* infection by hybrid therapy. It is also important to note that hybrid therapy consists of a dual regimen in the initial 7 days and a quadruple regimen in the final 7 days. Therefore, patients have to take two additional drugs in the last 7 days of therapy. The complicated drug administration may hinder the drug adherence of patients. Reversing the drug administration sequence of hybrid therapy (a quadruple regimen with a PPI, amoxicillin, clarithromycin and metronidazole in the initial 7 days followed by a dual regimen with a PPI and amoxicillin in the final 7 days) can simplify hybrid therapy and is beneficial for drug adherence [47]. A recent retrospective cohort study demonstrated that 14-day reverse hybrid therapy and 14-day hybrid therapy had comparable eradication rates [47]. Basically, reverse hybrid therapy is a modified 14-day concomitant therapy without clarithromycin and metronidazole in the final 7 days, and hybrid therapy is a modified 14-day concomitant therapy without clarithromycin and metronidazole in the initial 7 days.

PPIs possess two mechanisms of action in *H. pylori* eradication. They have direct anti-*H. pylori* activity, and they also can increase bioavailability and anti-*H. pylori* activity of antibiotics by increasing intragastric pH. In the first-line treatment of *H. pylori* infection, the type of PPI has been identified as an independent efficacy predictor of standard triple therapy [48]. A randomized controlled study showed that esomeprazole-based triple therapy demonstrated a higher eradication rate than a pantoprazole-based regimen [48]. The differences in eradiation efficacies between the two study groups may be related to the more powerful acid inhibition effect [49] and stronger anti-*H. pylori* activity of esomeprazole compared to pantoprazole [50,51]. A cross-over study showed that esomeprazole at a standard dose of 40 mg once daily provides more effective control of gastric acid at a steady state than standard doses of pantoprazole, lansoprazole and rabeprazole in patients with symptomatic gastroesophageal reflux disease [49]. Early studies suggested that esomeprazole might exert a stronger anti-*H. pylori* effect than pantoprazole. An in vitro study disclosed that omeprazole was more active against *H. pylori* than pantoprazole [50], and another study revealed lower MIC50 and MIC90 of esomeprazole compared to omeprazole [51]. In the current study, we also investigated the impacts of PPI types on the eradication outcome of hybrid therapy. The eradication rates of hybrid therapy based on esomeprazole, pantoprazole and rabeprazole were 99.1%, 93.1% and 94.3%, respectively. The differences in eradication rates did not reach statistical significance. The data suggest that the type of PPI dose not influence the efficacy of 14-day hybrid therapy.

In this study, *H. pylori* strains with (MIC) values >0.5 μg/mL, >1 μg/mL and >4 μg/mL were considered to be resistant to amoxicillin, clarithromycin and tetracycline, respectively. The criteria for the breakpoints of MIC values were different from those in the EUCAST (Clinical Breakpoint Tables v. 13.0, valid from 1 January 2023). The criteria of EUCAST are used frequently in European studies. The MIC breakpoints of amoxicillin, clarithromycin and tetracycline for *H. pylori* in EUCAST are 0.125 mg/L, 0.25 mg/L and 1 mg/L, respectively. It is important to note that MIC breakpoints identified in a specific treatment regimen may not be generalizable to other regimens. A recent study showed that the optimal MIC breakpoint for amoxicillin for antibiotic selection in the rescue treatment of *H. pylori* with PPI-bismuth-amoxicillin-tetracycline was >0.032 mg/L [52]. The optimal MIC breakpoints of various regimens for the first-line treatment of *H. pylori* merit further investigation.

There were some limitations in this study. First, this retrospective study was not a double-blind, placebo-controlled trial in which selection and assessment bias could be minimized. Second, there was only one patient harboring an amoxicillin-resistant strain. Therefore, the impact of amoxicillin resistance on the eradication outcome of hybrid therapy could not be assessed by the current study. Third, molecular testing for genetic resistances of *H. pylori* strains and host *CYP2C19* genotypes were unavailable. The strength of the study could be improved if it were possible to assess their relationships with eradication rates. Nonetheless, the current study is the largest study (*n* = 589) investigating the independent host and bacterial factors predicting eradication failure of 14-day hybrid therapy for the first-line treatment of *H. pylori* infection.

In conclusion, 14-day hybrid therapy achieves a high eradication rate for *H. pylori* infection in Taiwan. Clarithromycin resistance of *H. pylori* and poor drug adherence are independent risk factors predicting eradication failure of hybrid therapy.

## Figures and Tables

**Table 1 microorganisms-12-00006-t001:** Demographic data of the *H. pylori*-infected patients receiving 14-day hybrid therapy in the study.

Variables	Patient Characteristics (*n* = 589)
Age (yr) (mean ± SD)	54.5 ± 11.9
Gender (male/female)	271/318
Smoking	105/589 (17.8%)
Alcohol drinking	44/589 (7.5%)
Coffee consumption	133/589 (22.6%)
Ingestion of tea	181/589 (30.7%)
Endoscopic Findings	
Gastritis	373/589 (63.3%)
Peptic ulcer	219/589 (37.2%)
Antibiotic resistance	
Clarithromycin	54/350 (15.4%)
Amoxicillin	1/350 (0.3%)
Metronidazole	134/350 (38.3%)
Tetracycline	0/350 (0.0%)

**Table 2 microorganisms-12-00006-t002:** The treatment outcomes of the patients receiving 14-day hybrid therapy.

Variables	Outcomes
Eradication rate	
Intention-to-treat	548/589 (93.0%) 90.9–95.1% *
Modified intention-to-treat	542/574 (94.4%) 92.5–96.3%
Per-protocol	534/559 (95.5%) 93.8–97.2%
Adverse events	124/589 (21.1%) 17.8–24.4%
Drug adherence	575/589 (97.6%) 96.4–98.8%

* 95% confidence interval.

**Table 3 microorganisms-12-00006-t003:** Univariate analysis for factors affecting eradication rates of 14-day hybrid therapy in the first line treatment of *H. pylori* infection among the modified intention-to-treat population.

Characteristics	Patient Number	Eradication Rate	*p* Value
Sex			0.351
Male	266	255 (95.9%)	
Female	308	290 (94.2%)	
Age			0.175
<60 years old	327	314 (96.0%)	
≧60 years old	247	231 (93.5%)	
Cigarette smoking			0.939
Yes	102	97 (95.1%)	
No	472	448 (94.9%)	
Alcohol drinking			0.715
Yes	43	42 (97.75%)	
No	531	503 (94.7%)	
Coffee consumption			0.762
Yes	132	126 (95.4%)	
No	442	419 (94.8%)	
Tea consumption			0.204
Yes	180	174 (96.7%)	
No	394	371 (94.2%)	
Peptic ulcer			0.108
Yes	216	201 (93.1%)	
No	358	344 (96.1%)	
Antibiotic resistance			
Clarithoromycin			<0.001 *
Susceptible	287	280 (97.6%)	
Resistant	54	45 (83.3%)	
Amoxicillin			1.000
Susceptible	340	325 (95.6%)	
Resistant	1	1 (100%)	
Metronidazole resistance			0.935
Susceptible	208	199 (95.7%)	
Resistant	133	127 (95.5%)	
Tetracycline			—
Susceptible	341	326 (95.6%)	
Resistant	0	—	
Type and dose of PPI			0.064
Esomeprazole 40 mg bid	115	114 (99.1%)	
Pantoprazole 40 mg bid	162	151 (93.2%)	
Rabeprazole 20 mg bid	297	280 (94.3%)	
Drug adherence			0.005 *
Good	559	534 (95.5%)	
Poor	15	11 (73.3%)	

* denotes significant difference. (MIC breakpoint of clarithromycin: 1 μg/mL).

**Table 4 microorganisms-12-00006-t004:** Multivariate analysis for independent risk factors predicting eradication failure of 14-day hybrid therapy.

Risk Factors	Coefficient	Standard Error	Odds Ratio (95% CI)	*p*-Value
Clarithromycin resistance	1.585	0.609	4.8 (1.5–16.1)	0.009
Poor adherence	2.104	0.851	8.2 (1.5–43.5)	0.013

## Data Availability

The datasets generated and analyzed during the current study are not publicly available due to privacy or ethical restrictions but are available from the corresponding author on reasonable request.

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
