# Peer review of "Independent Risk Factors Predicting Eradication Failure of Hybrid Therapy for the First-Line Treatment of Helicobacter pylori Infection"

_microorganisms, 2023, doi:10.3390/microorganisms12010006_

Round 1

Reviewer 1 Report

Comments and Suggestions for Authors

This article deals with the H. pylori therapy and the relative eradication rates and failures. Many treatment models have been listed, beginning from the old standard triple therapy currently obsolete. The authors specifically  focus on the hybrid therapy containing a dual regimen with a PPI plus amoxicillin for initial 7 days followed by a quadruple regimen with a PPI plus amoxicillin, metronidazole and clarithromycin for final 7 days in a retrospective study concerning 589 patients in Taiwan.

This study is well conducted and well written leading to interesting results.  However there are some minor  flaws  to be addressed.  For instance the authors state that the comparison between this therapy and the classical one (Bismuth Quadruple Therapy BQT), shows that the hybrid therapy might have less adverse effects than BQT. However  this data is just  based  on the  literature  and not on the authors results.  Is this correct?  It should be more clearly specified. The study of the indipendent risk factors predicting  the eradication failure (such as CLA-resistance and drug adherence),  is explained in detail and in a correct way. The methods could be improved if it were possible to test the genetic resistance of Hp to clarithromycin.

In the title of the paragraphs 3.3 and 3.4 ,there are mistakes concerning the sentence  "bismuth quadruple therapy for H. pylori infectiony". In fact in the tables 3 and 4, the authors report their data concerning the hybrid therapy and not the bismuth quadruple therapy that they have never considered in the current study but only indirectly reported from literature. They should correct it.  Moreover the word "infectiony" should be replaced by the word "infection". The limitations of the article were appropriately considered by the authors.

Reviewer 2 Report

Comments and Suggestions for Authors

The topic of the manuscript is significant given the suboptimal eradication success of Helicobacter pylori infection over time. An advantage of the manuscript is the number (589 patients) evaluated. Although risk factors predicting H. pylori eradication failure are generally known, the benefits of the 14-day hybrid therapy provide an option for improving the therapy of the frequent, chronic and potentially tumorigenic infection. The work is well planned and carried out. I only have some suggestions that could enrich it.

 Introduction

1.       Lines 65-67: Seven-day therapy has no longer been the recommended option for treating H. pylori infection. This should be mentioned in the Introduction section.

2.       The authors should remove underlining in their manuscript.

3.       Line 112: The authors stated that the factors affecting the eradication efficacy of hybrid therapy remain unclear. In the review article of Hsu et al. (2015), clarithromycin resistance was mentioned to affect the success of hybrid therapy. (Hsu PI, Lin PC, Graham DY. Hybrid therapy for Helicobacter pylori infection: A systemic review and meta-analysis. World J Gastroenterol. 2015 Dec 7;21(45):12954-62. doi: 10.3748/wjg.v21.i45.12954.) Since Ping-I Hsu is a coauthor of this article as well as of the present work, this should be mentioned in the manuscript.

Methods

4.       Lines 148-149: Numbers of cases detected by culture and histology can be specified here.

5.       Lines 167-170: EUCAST resistance breakpoints (EUCAST Clinical Breakpoint Tables v. 13.0, valid from 2023-01-01) are different for amoxicillin, clarithromycin and tetracycline. They are used frequently, especially in European studies. It would be useful to mention and consider them in the results (including Table #3) and discussion as well.

Results

6.       Lines 196-197: see note #5.

7.       Line 220: correct “te” to “tea”.
